# Characteristics of the Uncinate Fasciculus and Cingulum in Patients with Mild Cognitive Impairment: Diffusion Tensor Tractography Study

**DOI:** 10.3390/brainsci9120377

**Published:** 2019-12-14

**Authors:** Chan-Hyuk Park, Su-Hong Kim, Han-Young Jung

**Affiliations:** Department of Physical and Rehabilitation Medicine, Inha University School of medicine, Inha University Hospital, Incheon 22332, Korea; chanhyuk@gmail.com (C.-H.P.); suhong1207@gmail.com (S.-H.K.)

**Keywords:** diffusion tensor tractography, mild cognitive impairment, uncinate, cingulum

## Abstract

Many studies have examined the relationship between cognition, and the cingulum and uncinate fasciculus (UF). In this study, diffusion tensor tractography (DTT) was used to investigate the correlation between fractional-anisotropy (FA) values and the number of fibers in the cingulum and UF in patients with and without cognitive impairment. The correlation between cognitive function, and the cingulum and UF was also investigated. Thirty patients (14 males, age = 70.68 ± 7.99 years) were divided into a control group (*n* = 14) and mild-cognitive-impairment (MCI) group (*n* = 16). The Seoul Neuropsychological Screening Battery (SNSB) and DTT were performed to assess cognition and bilateral tracts of the cingulum and UF. The relationship between SNSB values and the cingulum and UF was analyzed. The number of fibers in the right cingulum and right UF were significantly different between the two groups. The MCI group showed thinner tracts in both the cingulum and UF compared to the control group. A significant relationship was found between the number of fibers in the right UF and delayed memory recall. In conclusion, memory loss in MCI was associated with a decreased number of fibers in the right UF, while language and visuospatial function were related to the number of fibers in the right cingulum.

## 1. Introduction

The uncinate fasciculus (UF) connects the frontal and temporal lobes. The cingulum, being the largest fiber bundle, runs from the orbitofrontal cortex along the dorsal aspect of the corpus callosum to the temporal lobe [1,2]. The UF plays a role in emotion, memory, and language [1,3,4]. Conversely, the cingulum affects cognitive functions such as attention, memory, and motivation [5,6,7,8]. Although there have been many studies on the cingulum, the function of the UF is still unclear [5,6,9,10]. 

Diffusion tensor imaging (DTI) using a magnetic-resonance-imaging (MRI) technique provides information on the white-matter microstructure of the brain with regard to fiber connectivity and integrity [2]. Fractional anisotropy (FA) quantifies the preferential direction of water-molecule diffusion and gives an assessment of white-matter integrity [2]. It does this in the range of zero to one, where FA = zero is a sphere with perfect isotropic diffusion [11,12]. In patients with cognitive impairment due to diseases such as Alzheimer’s disease (AD) and traumatic brain injury, FA values for the cingulum and UF are decreased when measured by DTI [1,6,8].

There have been no studies in patients with cognitive impairment to examine the correlation between FA values and the number of fibers in the cingulum and UF, nor any that assess the more severely affected side of the brain. There have also been no studies to assess which tract has larger influence on cognitive deficits in these patients. Therefore, the aim of this study was to examine the correlation of FA values with the number of fibers in the cingulum and UF in patients without cognitive impairment and patients with mild cognitive impairment (MCI). In addition, this study investigated which tract had a larger effect on cognitive function, as well as the inter-relationship between the cingulum and UF. 

## 2. Materials and Methods

### 2.1. Subjects

Thirty patients (14 males, 16 females, age = 70.68 ± 7.99 (range 58–84) years) from Inha University Hospital who had complained of cognitive impairment were enrolled in the study. Patients with a history of stroke, traumatic brain injury, psychiatric disease, Parkinson’s disease, other neurological diseases, or an incomplete Seoul Neuropsychological Screening Battery (SNSB) were excluded. The patients were classified into 2 groups. The SNSB and a brain MRI were used to diagnose cognitive impairment. From the 30 patients, 14 patients with normal cognition were classified as controls (46.7%, 5 males, mean age = 66.00 ± 4.95 years). Sixteen patients diagnosed with MCI were allocated to the MCI group (53.3%, 9 males, mean age = 71.38 ± 8.61 years, Table 1). 

This study was performed retrospectively according to the Declaration of Helsinki. The study protocol was approved by the Inha University Hospital Institutional Review Board (INHAUH-2019-10-014).

### 2.2. MRI Acquisition

The neuroimaging and diffusion tensor tractography (DTT) in this study used a 3.0 T GE Signa Architect MRI system (General Electric, Milwaukee, WI, United States). The parameters for data acquisition were as follows: Repetition time (TR) = 13287 ms; echo time (TE) = 80.4 ms; field of view = 240 × 240 mm^2^; acquisition matrix = 128 × 128; b = 1000 mm^2^s^−1^; and slice thickness = 2 mm. Additional parameters for DTT were the direction of 30 and 72 contiguous slices. A rehabilitation doctor analyzed the DTT using DTI studio software (www.mristudio.org, Johns Hopkins Medical Institute, Baltimore, MD, United States). The reconstruction of the cingulum and uncinate area used 2 regions of interest (ROIs) in both the cingulum and UF [6,13]. In the cingulum, the first ROI was located in the middle part of the cingulum in the coronal plane. The second ROI was located in the posterior part of the cingulum, in the coronal plane on the right and left side [6]. Two ROIs in the UF were selected on a coronal slice along the most posterior plane of the temporal and frontal lobes. The first ROI was located in the entire temporal lobe on the right and left side. The second ROI was located in a small green zone in the same image [13]. The parameters for fiber tracking were an FA value of <0.15 or a turning angle of >60° [14]. The mean FA value and mean number of fibers in both the cingulum and UF were measured. 

### 2.3. Neuropsychological Tests

This study used the SNSB to assess cognitive impairment. The SNSB is a neuropsychological test battery for assessing the cognitive function of patients in Korea. It consists of 5 cognitive domains—attention, memory, language, visuospatial function, and frontal/executive function [15]. It is composed of the Korean Mini Mental State Examination (K-MMSE), the digit-span test to assess attention, a short form of the Boston Naming Test (short-BNT) to assess language, the Rey Complex Figure Test (RCFT) to assess visuospatial function, the Seoul Verbal Learning Test for the Elderly (SVLT-E) to assess memory, and the Go/No-Go test to assess frontal/executive function.

### 2.4. Statistical Analysis

SPSS software (version 22.0; SPSS, Chicago, IL, USA) was used for statistical analysis. Quantitative data represented the means ± standard deviations. One-way ANOVA and a Least Significant Difference (LDS) post hoc test were used to compare characteristics between groups. Correlation analysis was performed using Pearson’s correlation coefficient; *p* < 0.05 or < 0.01 were considered significant.

## 3. Results

Table 1 lists the SNSB results of the two groups for cognitive function. K-MMSE results showed significant difference between the control and MCI groups. Most SNSB components showed significant differences between the two groups (*p* < 0.01). However, there were no significant differences in the forward-digit-span score or frontal/executive function between the control and MCI groups (*p* > 0.05). 

Figure 1 and Figure 2 show evidence of thinner tracts in the MCI group than those in the control group in both the cingulum and UF. The neural tracts between cingula were found in the basal forebrain, travelling via the genu of the corpus callosum (Figure 1). FA values and the number of fibers in the MCI group were lower than those in the control group (Table 2). Although decreases in the number of fibers in the right cingulum and right UF were statistically significant, other parameters showed no significant differences (*p* < 0.05). FA values for the right cingulum showed significant correlation with the number of fibers in the right cingulum, FA values for the left cingulum, and FA values for the right UF. In addition, the number of fibers in the right cingulum was significantly correlated with the number of fibers in the left cingulum. Significant difference was not observed between FA values for bilateral UFs. To correlate all SNSB components with DTT reconstruction, we used Pearson’s correlation coefficient (Table 3). As a result, in the left UF, unlike the right UF, there was a relationship between the number of fibers and the FA values (Table 3, *p* < 0.05). While the number of fibers in the right cingulum was correlated with the short-BNT and RCFT scores, the number of fibers in the right UF was correlated with memory function, as measured by delayed recall. Other variables in the UF were not related to cognitive function. Age was only correlated with immediate memory recall (Table 3, *p* < 0.01).

## 4. Discussion

This study examined the relationship between cognitive function and DTT parameters in the cingulum and UF in patients with MCI compared to controls. Although the number of fibers and FA values for both the cingulum and UF decreased according to the severity of cognitive impairment, statistical significance was shown only for the number of fibers in the right cingulum and right UF (Table 2). On DTT, the MCI group showed a larger decrease in the posterior pathway and a lack of connection up to the basal forebrain in the right cingulum when compared to controls. UFs in the MCI group were bilaterally thinner than those in the control group (Figure 1 and Figure 2). There was a bidirectional relationship between FA values and the number of fibers in the cingula. All FA values for the UFs were related bilaterally, and the FA values for the left UF showed a correlation with the number of fibers in the left UF. While language (*p* = 0.384, *p* < 0.05) and visuospatial function (*p* = 0.384, *p* < 0.05) were related to the number of fibers in the right cingulum, correlation was only observed between the number of fibers in the right UF and delayed memory recall (*p* = 0.424, *p* < 0.05; Table 3). 

The number of fibers in the right UF and right cingulum affected cognition. The DTT of bilateral cingula in patients with MCI in this study showed evidence of discontinuation, meaning the forebrain ended prematurely. A thinner UF tract compared to the control group was also observed (Figure 1). Tracts in bilateral UFs in the MCI group were thinner than those in the control group (Figure 2). These results suggested a relationship between cognition, and the number of fibers in the right cingulum and right UF (Table 2). Our findings revealed that the right UF is correlated with the memory component of cognitive function (*p* = 0.424, *p* < 0.05), and that the right cingulum affects language (*p* = 0.384, *p* < 0.05) and visuospatial function (*p* = 0.378, *p* < 0.05). Cognitive impairment in MCI includes a deficit in the delayed recall of verbal memory and executive dysfunction [16,17]. Previous studies suggested a relationship between cognition, and the cingulum and UF [1,5,14,18,19]. In particular, the cingulum plays an important role in cognitive functions, such as memory [5,6]. The UF also reflects cognitive dysfunction in areas such as verbal memory, visual attention, verbal abstraction, categorization, and the immediate recall of word pairs [1]. These reports support our findings.

The region from the anterior cingulum to the forebrain has been associated with acetylcholine-mediated neurotransmission to the cerebral cortex, and a discontinuation of the cingulum can cause memory impairment [6,7]. The UF conveys cholinergic pathways to the origin of the basal nucleus of Meynert [1,19]. Neurons in the cholinergic basal forebrain are involved in synaptic plasticity, learning, memory, arousal, and attention. This means that the cholinergic pathway is essential for memory [20,21]. Given that the cholinergic pathway is related to memory, this study showed that a thinner UF tract in the cholinergic pathway of the forebrain results in cognitive impairment, resulting from longer distance from the basal forebrain (Figure 2). 

Unlike previous studies, this study failed to show any memory loss despite the discontinuation of the forebrain in the anterior cingulum [6,7]. This can be explained by the recovery of the cingulum, which was suggested in a previous study [22]. Jang and Seo described the recovery of the anterior cingulum after a brain injury. In particular, Mechanism 4 they suggested supports the present result, in that a neural tract from the contralateral basal forebrain, travelling via the genu of the corpus callosum, induced the recovery of the affected cingulum [22]. In addition, another study suggested that both the medial and frontal aspects of the cingulum retained connectivity [23]. This result supports the present study in which there was connectivity in bilateral frontal cingula, and the cingulum was not correlated with memory function (Figure 1 and Table 3). 

This study, as well as a previous study, showed that the DTT parameters of the UF were affected in MCI [1]. A previous study subdivided MCI patients into an early or late MCI group, and found that only the late MCI group showed correlation with the FA value for the UF [24]. By observing a significant reduction in SNSB results, and the number of fibers in the cingulum and UF between the two groups, this study showed a correlation between age, SNSB score, and DTT parameters in all subjects. This study found that the number of fibers in the right UF was significantly correlated with delayed memory recall (*p* = 0.424, *p* < 0.05). However, this finding is limited because there was no correlation found between FA values for the UF in the MCI and control groups. The reason for this may be that patients were not classified into early or late MCI groups, which was done in previous studies [24]. Further evaluation of this classification is necessary. 

This study found that cognitive impairment in MCI was related to a decrease in the number of fibers in the right cingulum and right UF. Language and visuospatial function were related to the number of fibers in the right cingulum, and memory was related to the number of fibers in the right UF (Table 3). It may be hypothesized on the basis of previous evidence that memory loss was not observed despite the decreased number of fibers in the right cingulum because its degeneration was compensated for by the left cingulum [22]. Only delayed memory recall showed a correlation with the number of fibers in the right UF. Therefore, because the DTT parameters of the right UF were not related to those of the left UF, this study suggests that delayed memory loss in MCI results from a decreased number of fibers in the right UF. The number of fibers in the right UF had an effect on memory loss in patients with MCI (Table 3). 

In the authors’ opinion, the number of fibers in the right UF was related to delayed memory recall in patients with MCI because the contralateral cingulum showed preserved connectivity. A limitation of this study is that the MCI group was not subdivided into early and late MCI groups due to the limited scale of the study. A larger-scale study, with patients divided into early and late MCI groups, is needed. Furthermore, repeated longitudinal studies are needed to analyze the current study’s participants in future years to confirm that the reported cognitive changes continued to evolve. Finding no significant difference in immediate memory recall requires further DTT-based research, given that a previous study described that immediate memory recall was correlated with the fornix [25]. We showed that immediate memory recall is only correlated with age. This result was supported by a previous study [26], where it was demonstrated that immediate memory recall declined with age. 

## 5. Conclusions

This study examined the relationship between UF and memory in patients with normal cognitive function and those with MCI. Memory loss in MCI is associated with the number of fibers in the right UF, and language and visuospatial function are related to the number of fibers in the right cingulum. We recommend that DTT is used in the confirmation and evaluation of cognitive impairment.

## Figures and Tables

**Figure 1 brainsci-09-00377-f001:**
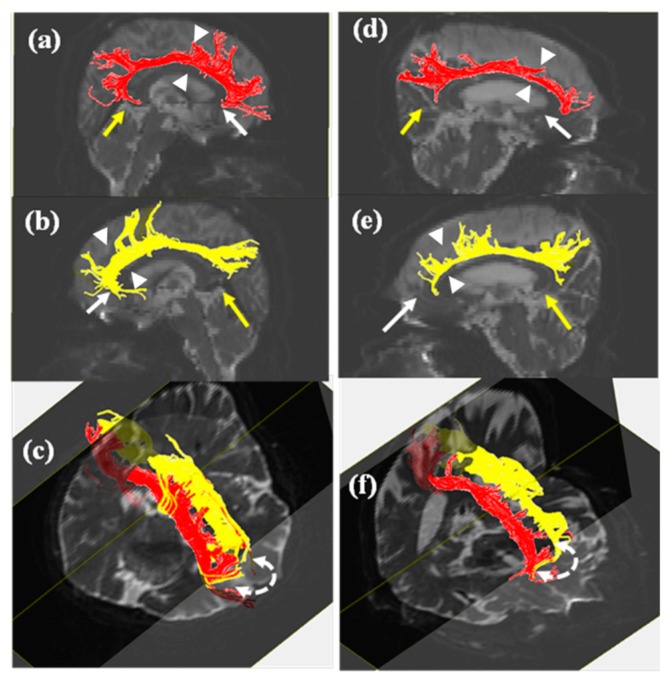
Diffusion tensor tractography (DTT) of right and left cingulum. (**a**) Left (red) and (**b**) right (yellow) frontal pathways in control group showed connection up to the basal forebrain. Activity in (**d**) left (red) and (**e**) right (yellow) frontal pathways decreased in MCI. Both posterior pathways in MCI ((**d**) left (red) and (**e**) right (yellow)) were thinner and shorter than those in control group ((**a**) left (red) and (**b**) right (yellow)). Changes in the structure of forebrain, anterior cingulate cortex, and posterior pathway in MCI shown. Fiber connectivity between bilateral cingula shown in (**c**) and (**f**). Note: white arrow-, anterior cingulate cortex; yellow arrow, posterior cingulate cortex; curved arrow, connectivity between right and left cingulum; and white arrow head, thinner region.

**Figure 2 brainsci-09-00377-f002:**
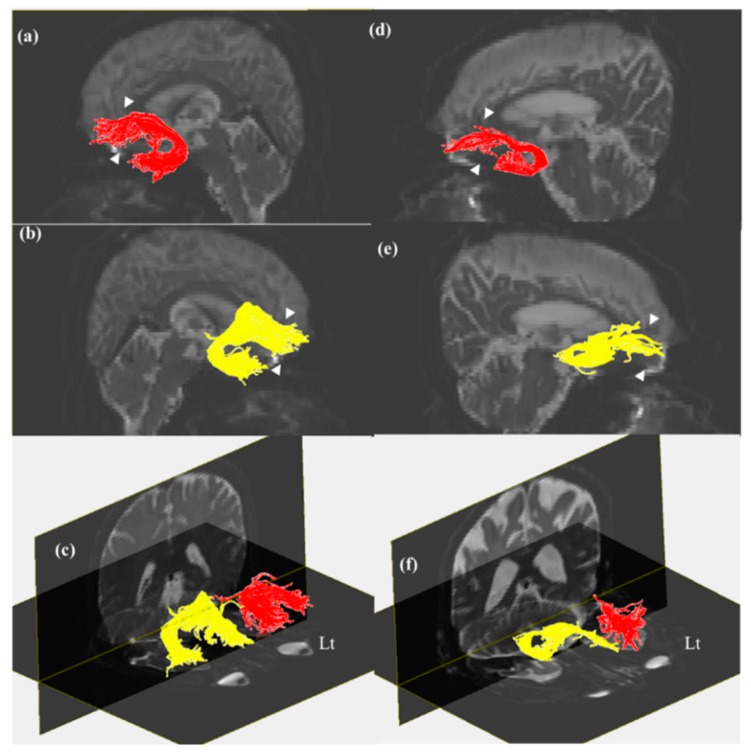
Diffusion tensor tractography (DTT) of right and left UFs. (**a**) Left UF in control group. (**b**) Right UF in control group. (**c**) Bilateral UFs in control group. (**d**) Left UF in MCI group. (**e**) Right UF in MCI group. (**f**) Bilateral UFs in MCI group. Tracts in MCI group were bilaterally thinner than those in control group (white arrow head, thinner region). Note: UF, uncinate fasciculus; and MCI, mild cognitive impairment.

**Table 1 brainsci-09-00377-t001:** Characteristics according to the severity of cognitive impairment. (* *P* < 0.01, ** *P* < 0.01).

	Control (*n* = 14, M/F: 5/9)		MCI (*n* = 16, M/F: 9/7)		*p*-Value
	Mean	SD	Mean	SD	
**Age, year**	66.00	4.95	71.38	8.61	0.044 *
**K-MMSE**	27.84	1.28	25.73	3.23	0.037 *
**Digit span**					
Forward	7.00	1.15	6.33	1.45	0.194
Backward	4.85	1.77	3.47	1.36	0.028 *
**Language**					
Short-BNT	12.46	1.50	9.73	2.60	0.003 **
**Visuospatial function**					
RCFT (copy score)	33.77	2.95	26.33	10.87	0.025 *
**Memory (SVLT-E)**					
Immediate recall	19.84	3.87	14.20	4.51	0.002 **
Delayed recall	6.46	1.45	2.00	2.32	0.000 **
**Frontal/Executive functions**					
Go-No-Go	19.38	0.96	16.33	5.61	0.065

Abbreviation: MCI: mild cognitive impairment, K-MMSE: Korean version mini-mental state examination, BNT: the boston naming test, RCFT: the Rey Complex Figure Test, SVLT-E: the Seoul verbal learning test for the elderly.

**Table 2 brainsci-09-00377-t002:** Comparison between the cognition assessments and DTT parameters. (* *P* < 0.01, ** *P* < 0.01).

		Control		MCI		*p*-Value
		Mean	SD	Mean	SD	
**FA of Cingulum**	Right	0.54	0.03	0.52	0.02	0.106
	Left	0.54	0.02	0.53	0.02	0.194
**FA of UF**	Right	0.48	0.03	0.47	0.03	0.639
	Left	0.48	0.02	0.47	0.045	0.723
**The number of fibers of Cingulum**	Right	1251.00	399.19	907.87	262.37	0.011 *
	Left	1217.85	289.94	983.07	332.48	0.059
**The number of fibers of UF**	Right	912.15	228.76	587.40	234.05	0.001 *
	Left	590.69	270.43	434.80	236.15	0.115

Abbreviation: MCI: mild cognitive impairement, UF: uncinate fasciculus, FA: fractional anisotropy, SD: standard deviation.

**Table 3 brainsci-09-00377-t003:** Pearson’s correlation between the cognition assessments and DTT parameters. (* *P* < 0.01, ** *P* < 0.01).

			Age	Right Cingulum		Left Cingulum		Right UF		Left UF	
				FA	The Number of Fibers	FA	The Number of Fibers	FA	The Number of Fibers	FA	The Number of Fibers
**Right Cingulum**	FA	*r*	−0.132	-							
	The number of fibers	*r*	−0.144	0.499 **	-						
**Left Cingulum**	FA	*r*	−0.285	0.671 **	0.366	-					
	The number of fibers	*r*	−0.126	0.301	0.415 **	0.236	-				
**Right UF**	FA	*r*	−0.066	0.434 *	0.236	0.453 **	−0.108	-			
	The number of fibers	*r*	0.193	0.065	0.295	−0.044	0.032	−0.089	-		
**Left UF**	FA	*r*	−0.230	0.370	0.219	0.387 **	0.111	0.647 **	−0.162	-	
	The number of fibers	*r*	−0.331	-0.191	0.141	0.124	−0.034	0.171	0.302	0.408 **	-
**Digit span**	Forward	*r*	−0.287	0.263	0.119	0.103	−0.174	−0.081	0.305	0.285	0.116
	Backward	*r*	−0.335	0.252	0.150	0.003	0.363	−0.190	0.325	0.211	0.169
**Language**	Short BNT	*r*	−0.359	0.194	0.384 **	0.013	0.230	−0.058	0.341	0.249	0.303
**Visuospatial function**	RCFT	*r*	−0.322	0.301	0.378 **	0.200	0.223	−0.014	0.249	0.269	0.096
**Memory (SVLT-E)**	Immediate recall	*r*	−0.539 **	0.155	−0.013	−0.034	0.223	−0.153	0.248	0.012	0.109
	Delayed recall	*r*	−0.145	0.144	0.161	0.031	0.207	−0.156	0.424 **	0.067	0.370
**Frontal/Executive function**	Go-no-Go	*r*	−0.329	0.369	0.324	0.174	0.181	−0.070	0.217	0.198	0.042

Abbreviation: UF: uncinate fasciculus, FA: fractional anisotropy, MCI: mild cognitive impairment, K-MMSE: Korean version mini-mental state examination, BNT: the boston naming test, RCFT: the Rey Complex Figure Test, SVLT-E: the Seoul verbal learning test for the elderly.

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
