# Peer review of "Characteristics of the Uncinate Fasciculus and Cingulum in Patients with Mild Cognitive Impairment: Diffusion Tensor Tractography Study"

_brainsci, 2019, doi:10.3390/brainsci9120377_

Round 1

Reviewer 1 Report

An interesting study evaluating changes in the cingulum and uncinate fasiculus in cognitively healthy individuals vs those with MCI . Some of the sentence constructions are hard to follow and require editing for clarity. The use of the term 'discontinuation of the forebrain' (line 142 page 6, line 169 page 7)in the discussion is undefined - do the authors mean the forebrain ended anatomically earlier than normal? The term 'progressing to MCI' is also an unusual language construct - I would suggest referring to specifically stating that controls & MCI were compared. 

Some of the conclusions are very definite and probably should be written more as hypotheses derived from the current results. English language review would greatly improve the paper. 

Suggestions for future work should include repeated longitudinal studies - it would be of great interest to repeat the studies in your participants in future years, to see if these changes further evolve as their cognition changes. 

Author Response

Point 1: An interesting study evaluating changes in the cingulum and uncinate fasiculus in cognitively healthy individuals vs those with MCI . Some of the sentence constructions are hard to follow and require editing for clarity. The use of the term 'discontinuation of the forebrain' (line 142 page 6, line 169 page 7)in the discussion is undefined - do the authors mean the forebrain ended anatomically earlier than normal?

Response 1: 

Line 142 Page6: we revised the term as the following; discontinuation of forebrain --> no connection up to the basal fore brain

Line170 Page 10

a discontinuation, which means the forebrain ended anatomically earlier, and a thinner tract than normal group (Figure 1).

Point 2: The term 'progressing to MCI' is also an unusual language construct - I would suggest referring to specifically stating that controls & MCI were compared. 

Response 2: We change ‘progressing to MCI’ to

Line 193 page 7: progressing to MCI --> in patients with MCI

Line 201 page 7: progressing to MCI -->in patients with normal cognition function and  MCI

Delete that “when processing from normal cognition to MCI (Page 10, 187)”

Point 3: Some of the conclusions are very definite and probably should be written more as hypotheses derived from the current results. English language review would greatly improve the paper. 

Response 3:

We revised conclusion, referring to your comment.

“The reason which cingulum was not correlated with memory is the connectivity in basal forebrain between both cingulum” was erased.

The conclusion was revised as the following

 “DTT plays an important role in predicting cognitive impairment. These results suggest that the DTT technique would be useful for confirming the cognitive impairment. DTT is recommended for evaluating cognition impairment” -->  “On the other hand, We recommend that because the DTT technique would be useful for confirming the cognitive impairment.  this is useful to evaluating cognition impairment.”

Point 4: Suggestions for future work should include repeated longitudinal studies - it would be of great interest to repeat the studies in your participants in future years, to see if these changes further evolve as their cognition changes. 

Response 4: 

Line 223 Page 11

We add 

“ On the other hand, for repeated longitudinal studies, through repeat of the studies in participants in future years, the confirmation that these changes further evolve their cognition changes will be necessary. “

Reviewer 2 Report

In this manuscript, the authors reported a DTT study showing that memory loss in MCI was associated with the number of fibers of the right uncinated fasciculus (UF), while language and visuospatial function was related to the number of fiber of the right cingulum. Overall, the conclusions are not fully supported by the presented data, and the critical information regarding the data analysis is missing.

Major:

Table 1, the average age of the patients with MCI was significantly older than that of the control. Would the age bias all the conclusions? The role of age in the cognition impairment should be at least discussed and correlated with all the parameters reported in this study. Figure 1, the illustrated DTT images of cingulum for the patients with MCI did not exhibit any sign of “discontinuation”. However, the authors stated that “a discontinuation…of the forebrain…” on Page 6, line 150 and Page 7, line 169. This needs to be clarified. In addition, the description for panels (a)-(f) needs to be provided. Figure 2, the panel number was mislabeled. It should be (d)-(f) other than (e)-(g) for the panels on the right. Table 2, there are no significant differences in FA values for either left or right UF when compared the patients with MCI to that of the normal control. It’s not clear why the authors stated that “a significant difference was observed between the FA values in both UFs (Page 3, lines 108-109)”. It’s not clear how the Pearson’s correlation was performed in Table 3. The denoted significances were obtained from the comparison between the patients and the control? The detailed information about the data analysis needs to be provided. Page 6, line 138, the authors stated that “…patients with normal cognition and without MCI”. It should be “patients with MCI and the control with normal cognition”.

Minor:

Page 2, lines 71 and 72, the full name for “OR” and “AND” needs to be provided.

Author Response

In this manuscript, the authors reported a DTT study showing that memory loss in MCI was associated with the number of fibers of the right uncinated fasciculus (UF), while language and visuospatial function was related to the number of fiber of the right cingulum. Overall, the conclusions are not fully supported by the presented data, and the critical information regarding the data analysis is missing.

Major:

Point 1: Table 1, the average age of the patients with MCI was significantly older than that of the control. Would the age bias all the conclusions? The role of age in the cognition impairment should be at least discussed and correlated with all the parameters reported in this study.

Response 1: 

We confirmed the correlation between age and all the parameters.

Line 112: We add the sentence, “The role of age in the cognition was only correlated with immediate recall memory (Table 3, P < 0.01)”and revised “Table 3”.

Line 202:

we add “However, we showed that immediate recall of memory was only correlated with age. This result was supported by the previous result [26]. They demonstrated that immediate recall of memory was declined according to age” and “ reference  “Arpawong, T.E.; Pendleton, N.; Mekli, K.; McArdle, J.J.; Gatz, M.; Armoskus, C.; Knowles, J.A.; Prescott, C.A. Genetic variants specific to aging-related verbal memory: Insights from GWASs in a population-based cohort. PLoS One 2017, 12, 1–27 “.

Point 2: Figure 1, the illustrated DTT images of cingulum for the patients with MCI did not exhibit any sign of “discontinuation”. However, the authors stated that “a discontinuation…of the forebrain…” on Page 6, line 150 and Page 7, line 169. This needs to be clarified.

Response 2:

Line 150 Page 6 and Line 169 Page 7.

We change that sentence to “DTT of both cingulum in patients with MCI in the present study showed a the forebrain ended anatomically earlier and a thinner tract than normal group discontinuation and a thinner tract of the forebrain in the both cingulum . and a thinner tracts in the both UFs of MCI group were thinner than that of the control.”

Point 3: In addition, the description for panels (a)-(f) needs to be provided. Figure 2, the panel number was mislabeled.

Response 3: We changed the panel numbers

 It should be (d)-(f) other than (e)-(g) for the panels on the right.

We revised the following:

Figure 1. Duffusion Tensor Tractograhy (DTT) of the right and left cingulum. The left (red, (a)) and right (yellow, (b)) frontal pathways in control group showed a connection up to the basal forebrain.  The left  (red, (d)) and right (yellow, (e)) frontal pathway decreased in MCI. The both posterior pathways in MCI (left: red (d), right: yellow (e)) was thinner and shorter than that in control (left: red (a), right: yellow (b)). According to the severity of cognitive impairment, the changes in the structures of the forebrain and anterior cingulate cortex, as well as the decreased posterior pathway are shown The connectivity of fibers between the cingulum was shown at two groups) ((c), (f)). (White arrow: anterior cingulate cortex, yellow arrow: posterior cingulate cortex curved arrow; connectivity between right and left cingulum, white arrow head: thinner region)

Point 4: Table 2, there are no significant differences in FA values for either left or right UF when compared the patients with MCI to that of the normal control. It’s not clear why the authors stated that “a significant difference was observed between the FA values in both UFs (Page 3, lines 108-109)”.

Response 4:

We add the term “not”.(Page 3, lines 108-109)

Point 5: It’s not clear how the Pearson’s correlation was performed in Table 3. The denoted significances were obtained from the comparison between the patients and the control? The detailed information about the data analysis needs to be provided.

Response 5:

In Page 3 line 109, we add the following sentence “

For correlation between all terms of SNSB and reconstruction of DTT, we performed Pearson’s coefficient (Table 3). As a result, unlike the right UF, the left UF revealed the relationship between the number of fibers and FA values”

In Page 8 line 199

We add the following sentence;

“Because the significant reduction in the number of fiber in cingulum and UF or terms of SNSB between two groups, the present study showed correlation between ages,  terms of SNSB, and parameters of DTT in all subjects. The result found that the number of fibers in the right UF was significantly correlated with delayed recall memory (P = 0.424, P < 0.05). However, this result is limitation in this study because of no correlation between FA value of UFs and MCI or control, respectively. Additionally, the reason is no classification as early and late MCI was different from the previous study [24].”

Point 6: Page 6, line 138, the authors stated that “…patients with normal cognition and without MCI”. It should be “patients with MCI and the control with normal cognition”.

Response 6:

Page 6 line 138

We revised the sentence as your comment " patients with MCI and the control with normal cognition”.

Point 7: Minor:

Page 2, lines 71 and 72, the full name for “OR” and “AND” needs to be provided. 

“OR” or “AND” is terms used when the reconstruction of DTT is performed

Response 7:

The terms as "AND" and "OR" were used when placing seed or target ROIs in DTT.

Please, refer to the following reference

Larroza, A.; Moratal, D.; D’ocón Alcañiz, V.; Arana, E. Tractography of the uncinate fasciculus and the posterior cingulate fasciculus in patients with mild cognitive impairment and Alzheimer disease. Neurol. 2014, 29, 11–20.

Reviewer 3 Report

The authors analyzed the diffusion tensor tractography (DTT) for thirty patients including the ones with mild cognitive impairment (MCI). They found a correlation with a decrease in the number of fibers in the right cingulum and uncinated fasciculus with memory. The study is represented clearly and concisely.  I suggest a few minor changes.

Explain all the abbreviations and terminologies in Table and figure legends again even if they have described them in the text.

In Line 91: it seems that authors mean ‘One-way ANOVA’ instead of One-way analysis.

Figure 1,2: Figure 1 does not say what is represented by a-f. For Figures 1 and 2, explain in detail what is shown in the figure, what are yellow and red colors. Figure 2 legend: Authors say ‘The bilateral tracts of MCI group were thinner than the control’, it would be more comprehensible if the authors say how much thinner.

Discussion can be improved by citing their figures and tables or the values/changes in the parameters when authors are describing and discussing the results.

Author Response

The authors analyzed the diffusion tensor tractography (DTT) for thirty patients including the ones with mild cognitive impairment (MCI). They found a correlation with a decrease in the number of fibers in the right cingulum and uncinated fasciculus with memory. The study is represented clearly and concisely.  I suggest a few minor changes.

Point 1: Explain all the abbreviations and terminologies in Table and figure legends again even if they have described them in the text.

Response 1: We added the abbreviations

Please, check all table or figures.

Point 2:

 In Line 91: it seems that authors mean ‘One-way ANOVA’ instead of One-way analysis.

Response 2: We change the terms as your comment 'One-way ANOVA'

Point 3:

Figure 1,2: Figure 1 does not say what is represented by a-f. For Figures 1 and 2, explain in detail what is shown in the figure, what are yellow and red colors. Figure 2 legend: Authors say ‘The bilateral tracts of MCI group were thinner than the control’, it would be more comprehensible if the authors say how much thinner.

Response 3: We demonstrated colors in Fig 1,2, and add the thinner region (white arrow head)

a-f in Figure 1,2 was represented.

The following sentences are revised sentences.

Figure 1. Duffusion Tensor Tractograhy (DTT) of the right and left cingulum. The left (red, (a)) and right (yellow, (b)) frontal pathways in control group showed a connection up to the basal forebrain.  The left  (red, (d)) and right (yellow, (e)) frontal pathway decreased in MCI. The both posterior pathways in MCI (left: red (d), right: yellow (e)) was thinner and shorter than that in control (left: red (a), right: yellow (b)). According to the severity of cognitive impairment, the changes in the structures of the forebrain and anterior cingulate cortex, as well as the decreased posterior pathway are shown The connectivity of fibers between the cingulum was shown at two groups) ((c), (f)). (White arrow: anterior cingulate cortex, yellow arrow: posterior cingulate cortex curved arrow; connectivity between right and left cingulum, white arrow head: thinner region)

Figure 2. Duffusion Tensor Tractograhy (DTT) of the right and left UFs. (a) Left UF of control, (b) Right UF of control, (c) Both UFs of control, (d) Left UF of MCI group, (e) Right UF of MCI group, and (f) Both UFs of MCI group. The bilateral tracts of MCI group were thinner than the control (white arrow head: thinner region). UF: uncinate fasciculus, MCI: mild cognitive impairment

Point 4:

Discussion can be improved by citing their figures and tables or the values/changes in the parameters when authors are describing and discussing the results.

Response 4:

Line 171, 173 Page 10: add (Figure 1) and (Figure 2)

Line 189 Page 10: add (Figure 2).

Line 198 Page 10: add (Figure 1, and Table 3)

Line 161 Page 10: add (Table 2)

Line 163 Page 10: add (Figure 1 and 2)

Line 176 Page 10: add (Table 2

Line 167,168, 170 Page 10: add the values

Round 2

Reviewer 2 Report

All the concerns raised in the previous review were addressed in the revised manuscript. Overall, the revised manuscript has been strengthened and improved.